# First-Line Immunotherapy with Check-Point Inhibitors: Prospective Assessment of Cognitive Function

**DOI:** 10.3390/cancers15051615

**Published:** 2023-03-06

**Authors:** Jamie S. Myers, Adam C. Parks, Jonathan D. Mahnken, Kate J. Young, Harsh B. Pathak, Rajni V. Puri, Amber Unrein, Phyllis Switzer, Yazan Abdulateef, Samantha Sullivan, John F. Walker, David Streeter, Jeffrey M. Burns

**Affiliations:** 1School of Nursing, University of Kansas, Kansas City, KS 66160, USA; 2Department of Neurology, University of Kansas Medical Center, Kansas City, KS 66160, USA; 3Department of Biostatistics & Data Science, University of Kansas Medical Center, Kansas City, KS 66160, USA; 4Department of Pathology and Laboratory Medicine, University of Kansas Medical Center, Kansas City, KS 66160, USA; 5University of Kansas Alzheimer’s Disease Research Center, Fairway, KS 66111, USA; 6Department of Quality Assurance, University of Kansas Medical Center, Kansas City, KS 66160, USA

**Keywords:** checkpoint inhibitors, immunotherapy, cognitive function, cancer, first-line therapy

## Abstract

**Simple Summary:**

About 40% of people diagnosed with cancer are eligible for treatment with checkpoint inhibitors (CPIs). Little research has been conducted to understand whether CPIs affect cognitive function. Most research that has been conducted included patients who received both CPIs and chemotherapy. This pilot study was conducted with patients receiving CPIs as their first line of cancer treatment. This study’s purpose was to show the feasibility of recruiting, retaining, and assessing older adult patients with cancer starting first line CPI treatment and provide early evidence of the impact of CPI on cognitive function.

**Abstract:**

Approximately 40% of patients with cancer are eligible for check-point inhibitor (CPI) therapy. Little research has examined the potential cognitive impact of CPIs. First-line CPI therapy offers a unique research opportunity without chemotherapy-related confounders. The purpose of this prospective, observational pilot was to (1) demonstrate the feasibility of prospective recruitment, retention, and neurocognitive assessment for older adults receiving first-line CPI(s) and (2) provide preliminary evidence of changes in cognitive function associated with CPI(s). Patients receiving first-line CPI(s) (CPI Group) were assessed at baseline (*n* = 20) and 6 months (*n* = 13) for self-report of cognitive function and neurocognitive test performance. Results were compared to age-matched controls without cognitive impairment assessed annually by the Alzheimer’s Disease Research Center (ADRC). Plasma biomarkers were measured at baseline and 6 months for the CPI Group. Estimated differences for CPI Group scores prior to initiating CPIs (baseline) trended to lower performance on the Montreal Cognitive Assessment-Blind (MOCA-Blind) test compared to the ADRC controls (*p* = 0.066). Controlling for age, the CPI Group’s 6-months MOCA-Blind performance was lower than the ADRC control group’s 12-months performance (*p* = 0.011). No significant differences in biomarkers were detected between baseline and 6 months, although significant correlations were noted for biomarker change and cognitive performance at 6 months. IFNγ, IL-1β, IL-2, FGF2, and VEGF were inversely associated with Craft Story Recall performance (*p* < 0.05), e.g., higher levels correlated with poorer memory performance. Higher IGF-1 and VEGF correlated with better letter-number sequencing and digit-span backwards performance, respectively. Unexpected inverse correlation was noted between IL-1α and Oral Trail-Making Test B completion time. CPI(s) may have a negative impact on some neurocognitive domains and warrant further investigation. A multi-site study design may be crucial to fully powering prospective investigation of the cognitive impact of CPIs. Establishment of a multi-site observational registry from collaborating cancer centers and ADRCs is recommended.

## 1. Introduction

The American Cancer Society estimates 1.9 million new diagnoses of cancer for 2022 [1]. The rates of cancer and cancer-treatment-related cognitive impairment (CRCI) for survivors of non-central nervous system (CNS) malignancies ranges as high as 75%, with estimates of 30% of survivors experiencing long-lasting cognitive impairment [2,3,4,5]. CRCI for non-CNS malignancies has a significant impact on a cancer survivor’s quality of life at home, at work, and socially [6,7,8,9,10,11]. Commonly reported issues of CRCI include trouble within the cognitive domains of short-term memory, attention and concentration, executive function, and visuospatial ability [2,12,13]. These issues translate into difficulties with word finding, reading complex material, forgetting appointments, misplacing items, and, in some cases, issues with driving [9,14,15]. Given the prevalence of these issues and the number of individuals who experience them, enhanced understanding of the treatments that contribute to CRCI is crucial for informed consent prior to administration of therapies, accurate patient/family education, and on-going assessment. Understanding the mechanisms involved in the development of CRCI is critical to the investigation of effective interventions.

Evidence within the CRCI literature supports the association between prolonged production of inflammatory cytokines and the cognitive changes attributed to diagnosis and treatment of non-CNS malignancies [16,17]. Specifically, the body’s response to malignant cells and various cancer treatments, such as chemotherapy, induces the expression of a number of cytokines in the peripheral blood. Cytokines are thought to both actively and passively cross the blood–brain barrier and stimulate further cytokine release within the central nervous system inflammatory network. Cytokines are proposed to play a role in neuroprogenitor cell injury through interaction with cytokine-specific receptors in neuronal or endothelial brain cells [18]. Cytokines of interest include: Interleukin-1 alpha, (IL-1α), IL-1ß, IL-2, IL-6, tumor necrosis factor alpha (TNFα), and interferon beta (INFß). Animal models of neuroinflammation in the context of cancer and treatment with radiation therapy and immunotherapy indicate changes also may occur in interferon gamma (IFNγ) and fibroblast growth factor-basic (bFGF) levels [19]. Neurotrophic factors (such as brain derived neurotrophic factor-BDNF), and other growth factors involved with neurogenesis (such as insulin-like growth factor 1 (IGF1) and vascular endothelial growth factor (VEGF)) also are of interest in preparation for future investigations of effective interventions to improve cognitive function [20,21,22,23,24,25].

The inflammatory response to the cancer and cancer therapy has been postulated to have both a direct and indirect effect on neuroprogenitor cells, functional and structural connectivity, and cognitive function. Release of pro-inflammatory cytokines within the central nervous system and peripheral blood, and down-regulation of brain-derived neurotrophic factor are associated with CRCI. Other candidate pathways include biological pathways common to aging (e.g., hormonal changes in estrogen/testosterone, damage to DNA repair mechanisms, shortening of telomeres, and reduction of brain blood flow). Cancer and cancer therapy are postulated to accelerate cognitive aging. Thus, older adults treated for cancer may be at greater risk for cognitive impairment.

At the time this study was being developed, the literature indicated that recent advances in immunotherapy with checkpoint inhibition had resulted in approvals for 7 drugs with indications for more than 15 tumor types [26]. Initially, checkpoint inhibitors (CPIs) only were given as second-line or later therapies. However, recent approvals have been granted for first-line and combination regimens with both immunotherapy and chemotherapy or other targeted agents in addition to later-line therapies following recurrence after chemotherapy [27,28,29]. CPIs have been developed to inhibit programmed cell death protein (PD-1), programmed death ligand (PD-L1), and cytotoxic T-lymphocyte antigen (CTLA-4) [30]. Combination regimens with more than one CPI are able to target more than one receptor on the T cells. Other approved combined regimens include inhibitors of VEGF and various chemotherapy agents. The mechanism of action for checkpoint inhibition involves stimulation of an immune response to the cancer that results in the expression of inflammatory products, including cytokines. Immune-related adverse events (irAEs) are attributed to this inflammatory response [26,30,31]. More severe irAEs have been associated with CTLA-4 inhibition and combination regimens [32,33]. To date, scant research has been conducted to determine the impact of immunotherapy on cognitive function [34]. What little research has been conducted has primarily involved patients who previously had received chemotherapy, making teasing out the cognitive impact for multiple lines of therapy difficult [34]. First-line treatment with CPIs provides a unique and compelling opportunity to study the impact of this form of immunotherapy alone (i.e., without confounding by other cancer treatment) on cognitive function. A recent systematic review indicated that over 40% of patients with cancer now are considered eligible for CPI therapy, and this number will continue to increase [35]. In addition to intravenous infusions investigation, the development of oral formulations of these drugs is also underway. Development of a prospective, observational registry for individuals receiving first-line checkpoint inhibition for cancer would contribute important information to the state of the science about the potential impact of checkpoint inhibition on cognitive function for a growing population.

The purpose of this prospective, observational pilot study was to (1) demonstrate the feasibility of prospective recruitment, retention, and neurocognitive assessment for older adults with non-CNS malignancies receiving first-line treatment with CPI(s) and (2) provide preliminary evidence of changes in cognitive function associated with checkpoint inhibitor immunotherapy. Comparisons between baseline and 6-month assessments for changes in self-report of cognitive function and performance on neurocognitive tests were planned to generate an effect size to inform future prospective research with an observational registry.

The pilot study objectives were to:Aim 1: Demonstrate the feasibility of recruiting, assessing, and retaining 20 older adults (>/= age 60) newly diagnosed with cancer who will receive first-line therapy with CPI(s) (CPI Group);Aim 2a: Estimate change and variability in participants’ self-reports of cognitive function and objectively measured neurocognitive performance over time: Baseline (T1: within 1–2 weeks of initiation therapy with CPIs) and 6 months later (T2);Aim 2b: Estimate change and variability in inflammatory and neurotrophic biomarkers between T1 and T2;Aim 2c: Compare change and variability in CPI Group participants’ objectively measured neurocognitive performances between T1 and T2 to existing control data available from the University of Kansas Alzheimer’s Disease Research Center (ADRC) database for age-matched cognitively intact cohort participants (data recorded at baseline and 12 months).

Pilot outcomes were expected to firmly establish successful recruitment procedures and identify potential barriers to retention. Information also was anticipated regarding the evolution of tumor types most likely to be represented within the institutional catchment area in addition to the previous projections based on retrospective Tumor Registry data.

## 2. Materials and Methods

### 2.1. Collaboration

Alzheimer’s Disease Research Centers across the United States contribute data to the National Alzheimer’s Coordinating Center (NACC) Uniform Data Set (UDS). NACC has developed a standardized neuropsychologic battery to test the cognitive domains of executive function, episodic memory, attention/working memory, and language/semantic memory. ADRCs administer this standardized battery to hundreds of participants annually who have provided informed consent for the use of their de-identified data for research purposes. The UDS participants cover a broad range of neuropsychological performance, including a robust population of cognitively intact participants. As of December 2022, the UDS includes annual neuropsychological data for over 11,000 cognitively intact participants aged 35–84 years and over 3000 who are under age 65 (UDS Demographics and diagnoses | National Alzheimer’s Coordinating Center (naccdata.org, accessed on 5 February 2023). The study design and implementation for this pilot study resulted from a collaborative effort between the National Cancer Institute-designated Comprehensive Cancer Center (University of Kansas Cancer Center-KUCC) and the National Institute on Aging funded by the Alzheimer’s Disease Research Center (University of Kansas Medical Center-KUMC ADRC). The annual neuropsychological assessments are administered by the KUMC ADRC psychometricians. This collaboration leveraged the ADRC infrastructure for standardized neurocognitive assessment and provided the age-matched control data from the pool of cognitively intact cohort patients used for control comparisons in this pilot study.

### 2.2. Eligibility

Control data were included from cognitively intact adults aged 60 and older with a Clinical Dementia Rating Score of zero and without clinically meaningful deficits in their cognitive performance. Control data were excluded for individuals with clinically meaningful depression or anxiety, Parkinson’s disease, cancer within the last 5 years (except non-metastatic basal or squamous cell carcinoma), history of drug or alcohol abuse (DSM-IV criteria) in the last 2 years, and visual or auditory limitations or other systemic or neurological disease that may interfere with cognition. The eligible pool included 231 participants who were cognitively intact at baseline and at the 12-month follow up.

The most recent fully extracted 12-month Tumor Registry data were used to estimate the potential pool of patients receiving first-line therapy with CPI(s). These data indicated that approximately 200 patients would meet eligibility criteria. Initial inclusion criteria required participants to be age 60 or older, scheduled to receive first-line treatment with CPI(s) (combination therapy with more than one CPI was accepted), diagnosed with any stage of non-CNS malignancy (without brain metastases), chemotherapy naïve, and able to speak and read English. Patients initially were excluded for previous receipt of a BRAF or tyrosine kinase inhibitor, or comorbidities affecting cognitive function (such as Alzheimer’s Disease or related dementias). Participation in other clinical trials was not automatically exclusionary but evaluated on a case basis by the research team. Concomitant participation in the institutional Biospecimen Repository Core Facility (BRCF) protocol was required, allowing the collection, storage, and analyses of extra blood sampling at the time of standard of care lab sampling.

### 2.3. Recruitment

Recruitment was planned from the five KUCC sites located within the Kansas City metro area. Potential participants were to be identified by medical oncologists, urologists, nurse practitioners, and other advanced practice providers at the time of diagnosis and treatment planning. Clinical research coordinators were assigned to the study to assist with identification of eligible individuals. The principal investigator (PI) presented the study at all pertinent tumor-specific disease working groups, and the study synopsis and recruitment flyers were provided to all providers.

### 2.4. Data Collection

Informed consent for this pilot was obtained from all CPI Group participants. CPI Group participants were asked to complete the study questionnaires (see Instruments, below) at baseline (within 1–2 weeks of initiating CPI therapy) and six months later during regularly scheduled clinic visits. CPI Group participants were also asked to complete a neurocognitive assessment within the same timeframes. The neurocognitive assessment was administered by the KUMC ADRC psychometricians under the oversight of the study team’s clinical neuropsychologist. The neurocognitive tests were congruent with the annual battery (baseline and 12 months) administered to the longitudinal ADRC cohort of cognitively intact controls for the NACC UDS and are outlined below (see Table 1).

### 2.5. Lab Sampling and Processing

The KUCC BRCF protocol permits collection of up to 6 tubes of blood from each participant at each visit. Lab sampling occurred during scheduled venipuncture for standard of care lab sampling in conjunction with participants’ CPI treatment at baseline and six months. These samples were de-identified (marked only with participants’ study identification numbers) and stored until samples from all timepoints from all participants were collected. At that time, the KUCC Biomarker Discovery Lab (BDL) staff obtained 1 mL of the deidentified plasma baseline and 6-month samples to quantify circulating levels of the study biomarkers using Luminex assays (Millipore-Sigma, Temecula, CA, USA) analyzed on a BioPlex 200 instrument. The mean-fluorescent intensity data generated by the Luminex assays for each marker was compared to standard curves and the resulting concentrations were supplied to the biostatistician for the analyses described below.

### 2.6. Self-Report Instruments

CPI Group participants’ self-reports of cognitive issues were measured with two instruments developed and validated by the National Institutes of Health [36]. The Patient Reported Outcomes Management Information System (PROMIS) Cognitive Function and Cognitive Function Abilities 8-item short forms were developed to measure problems with cognitive function and the perception of cognitive ability, respectively. Items from these instruments are ranked from 1 to 5. T-scores are calculated and used for continuous variable comparisons. Potential confounders associated with changes in cognitive function include depression, anxiety, activity level, and sleep quality. Well-validated, psychometrically sound instruments to measure these potential confounders were included and outlined in Table 2. The National Alzheimer’s Coordinating Center (NACC) Functional Assessment Scale also was administered to evaluate the potential impact of cognitive function on activities of daily living.

### 2.7. Neurocognitive Assessment

An abbreviated 60 min neurocognitive assessment battery was selected from the NACC standardized neurocognitive battery administered annually to the KUMC ADRC cognitively intact cohort. All NACC tests were planned for inclusion with the exception of the Benson Complex Figure Copy and Multilingual Naming Test. This battery was further supplemented with four additional planned tests routinely administered as components of the KUMC-specific ADRC standard battery (Digit Symbol, Block Design, Stroop Test, and Letter Number Sequencing). These selections were made to specifically focus on the cognitive domains of attention/concentration, working memory, processing speed, and visuospatial ability (Table 1), and to minimize redundancy and participant burden.

### 2.8. Data Analyses

To estimate the changes in self-report of cognitive function, performance on neurocognitive tests, and inflammatory markers, difference scores were calculated for each participant. Within-group changes in self-report of cognitive function and inflammatory markers were analyzed for the CPI Group participants. Between-group changes in performance on neurocognitive tests were analyzed for the CPI Group participants and the ADRC cognitively intact cohort. Mean difference scores were estimated, along with corresponding 95% confidence intervals. Pearson’s correlation coefficients between mean difference scores for the biomarker levels and cognitive variables were examined for the CPI Group participants, or, if indicated, Spearman’s correlation coefficients were used as a nonparametric alternative. For comparison to controls, linear mixed models were utilized to account for repeated measures comparing recruited versus control participants with data available from the KUMC ADRC cohort. Repeated measures using age as the time covariate enabled comparison in these measures with decline adjusted for age and allowed for comparison in slope between the recruited versus the control population to estimate and test for differences. Residual analyses were conducted, and nonparametric methods were utilized, if indicated, in place of the parametric approaches described.

### 2.9. Study and Recruitment Procedure Modification

The pilot study was opened in September of 2020. Recruitment challenges were experienced due to a number of factors, including: reduction of in-person clinic visits due to the on-going COVID-19 pandemic, significant turnover in clinical research coordinators supporting the study, limitation to one KUCC recruitment site due to restrictions in clinical research coordinator coverage, and the prevalence of CPI therapy administered in conjunction with chemotherapy agents. To address these challenges, the eligibility criteria for the study were revised to remove the age-based restrictions and to allow previous therapy with a BRAF or tyrosine kinase inhibitor. Administration of the neurocognitive assessments was shifted from in-person to phone. This shift resulted in omission of three of the planned neurocognitive assessments: digit symbol, block design, and Stroop, and included use of the phone version of the Montreal Cognitive Assessment (MoCA-Blind) and Trail Making Tests without tasks requiring visual abilities). Total scores on the MoCA-Blind range from 0 to 22 instead of from 0 to 30 on the in-person test. The Oral Trail Making Tests require the participant to orally respond with sequential numbers (1–25, Test A) and alternating sequential numbers and letters (1–12, A-L, Test B). Scoring is based on the time in seconds required to complete the test and the number of correct responses. Despite the fact that the NACC had developed a phone version for the standardized neurocognitive battery, the KUMC ADRC implementation had not yet yielded 12-month phone assessments for the cognitively intact cohort. Thus, only the MoCA Blind assessment was available for between-group analyses. In March of 2021, the use of the Curated Clinical Outcomes Database (C3OD) was created by the University of Kansas Medical Center Department of Biostatistics and Data Science to facilitate translational cancer research in the patient population by combining disparate, complex data sources into a single, curated, referential source that is updated daily. C3OD data pulls were instituted to identify eligible patients based on the revised study outcome criteria. Refinements were made to the weekly data pulls provided to the clinical research coordinators and PI in May of 2021 and remained active throughout the remainder of study recruitment.

## 3. Results

### 3.1. Sample

The cognitively intact control data were utilized for participants who had a documented MoCA-Blind score for at least two consecutive visits (about 12 months apart). To better align with the CPI Group demographics, patients older than 90 years at their baseline visit and those with more than 20 years of education were excluded.

CPI Group participant recruitment was initiated in September of 2020. The first participant consented on 9/11/20 but was hospitalized prior to baseline assessment and not included in the analyses. No further participants were recruited until after the study modifications were approved in November of 2020. Upon institution and refinement of the C3OD database pulls, 21 participants were recruited between May and December 2021, bringing the number available for baseline assessments to 20 (see Figure 1). Data collections for the 6-month assessments were completed by May of 2022. The majority of the cancer patient participants were diagnosed with melanoma (*n* = 12) (see Table 3). The remaining tumor types included renal cell, head and neck, urothelial, hepatocellular, and non-small cell lung cancers. Most participants received either pembrolizumab (*n* = 7, 33%) or nivolumab (*n* = 6, 28.5%). The remainder received combination therapy with either nivolumab/ipilumumab, pembrolizumab/axitinib, or atezolizumab/bevacizumab. The CPI Group participants were primarily white (95%), males (60%) with a mean of about 15 years of education (range 12–20 years) (Table 3). Little difference was noted between the cancer patient participants (at 6 months) and the ADRC controls (at 12 months) with the exception of age ranges (Table 4). Four of the CPI Group participants’ ages were 56 or younger (32, 34, 43, 56 years). However, the mean age for both groups was >65.

### 3.2. Self-Report Instruments

Descriptive statistics for the self-report instruments are listed in Appendix A (Table A1 and Table A2). No significant within-group changes were noted for the self-report instruments completed by the CPI Group participants (Table 5 and Table 6). Notably, CPI Group participants’ scores for the PROMIS Cognitive Function and Cognitive Abilities 8-item short forms both decreased by 3 points.

### 3.3. Neurocognitive Tests

Descriptive statistics for CPI Group participant scores on the cognitive tests are listed in Appendix A (Table A3). No significant within-group changes were noted for CPI Group participants’ performances on the neurocognitive tests (Table 7). Comparisons of the MoCA-Blind test scores for both groups are depicted in Figure 2 and Table 8. Estimates for differences in MoCA-Blind scores after receiving CPI treatment (6-month assessments) compared to baseline (within 1–2 weeks of initiating CPI treatment) were not significant (*p* = 0.277). However, estimated differences for CPI Group participants’ MoCA-Blind test scores compared to the ADRC cognitively intact controls approached significance at baseline (*p* = 0.066) and were significantly worse than the controls’ scores after CPI treatment (6 months for CPI Group participants’ scores compared to controls’ scores at 12 months, *p* = 0.011). The trajectory of scores for the CPI Group participants was lower than that of the controls for participants of all ages.

Given the fact that most CPI Group participants were diagnosed with melanoma (*n* = 12, 60%), we also conducted post facto analyses for this subgroup. Results mirrored that of the full CPI Group sample in that estimates for differences in MoCA-Blind scores after receiving CPI treatment compared to baseline were not significant (*p* = 0.2333). Estimated differences for the subgroup compared to the ADRC cognitively intact controls approached significance at baseline (*p* = 0.091) and were significantly worse than controls’ scores at 12 months (*p* = 0.013).

### 3.4. Biomarkers

Descriptive statistics for biomarker levels are depicted in Appendix A (Table A3). Scheduling issues prevented 6-month sampling for two participants. However, 3-month data were available for both and were included in the analyses. No significant change from baseline was noted for any of the biomarkers assessed (Table 9). Correlations for change from baseline between biomarkers and neurocognitive test scores are depicted in Table 10. Some significant correlations were noted. Inverse correlations were noted between IFNγ, IL-1β, IL-2, and FGF2 and performance on the Craft story recall. Unexpected inverse correlation was noted between IL-1α and the total number of seconds needed for completion of Oral Trail Making Test B (lower time equals better performance). Positive correlation was noted between IGF-1 and the letter-number sequencing test performance. Positive correlation was noted between VEGF and digit-span backwards performance. However, inverse correlation was noted between VEGF and Craft story recall performance.

## 4. Discussion

This pilot study yielded important information regarding the feasibility of investigating the potential cognitive impact of checkpoint inhibitor treatment. A number of challenges were experienced due to the ongoing pandemic, such as study staff attrition and reluctance of participants to attend in-person assessments. Recruitment was further complicated by limitations in study team availability to obtain consent from participants in all five of the KUCC clinical sites. Likewise, a majority of the participants screened for the cancer center site covered by the clinical research coordinators were not eligible due to planned combination therapy with chemotherapy. As a result, protocol modifications were made to relax the age restrictions and requirement for in-person study assessments. Implementation of the C30D database pulls for identification of eligible participants markedly enhanced the success of recruitment efforts.

Despite the age disparity between the CPI Group participants (mean age 66) and the ADRC cognitively intact controls (mean age 80), and the fact that only the MoCA-Blind scores were available to compare between groups, a significant difference was estimated at the second assessment timepoint for patients who had received CPI treatment.

Some significant correlations were demonstrated between various biomarkers and performance on neurocognitive tests. Most of these were in the expected direction, in that biomarkers known to be pro-inflammatory (IFNγ, IL-1β, IL-2, and FGF2) were inversely correlated with neurocognitive performance. However, the inverse correlation between IL-1α and Oral Trail Making Test B performance was surprising. Likewise, we anticipated that VEGF levels would be positively correlated with neurocognitive performance given the known role VEGF plays in promotion of vascular endothelial cells and proliferation of neuronal precursors. One exception to this was the inverse correlation between VEGF and performance on the Craft Story Recall. A potential explanation may be that VEGF levels are known to increase in association with an inflammatory response due to its role in angiogenesis and vascular permeability. Up-regulation of VEGF is noted in conjunction with cytokine expression [37].

Findings from this small pilot must be considered to be very preliminary, and a number of limitations must be acknowledged. Given available funding and the timing for research support, blood samples only were collected and analyzed for the CPI group, not allowing longitudinal comparisons for biomarker levels between the CPI Group and the cognitively intact controls. Neuropsychological testing was conducted at baseline and six-months for the CPI Group and compared to available data from baseline and twelve-month assessment for controls. The pilot study was not able to stratify results by tumor type, which may be an important future consideration due to the potential for variability in levels of cytokine production. In addition, the pilot study was not able to stratify by type of CPI regimen. As noted earlier, irAEs are more severe for patients receiving anti-CTLA-4 CPIs and for those receiving combination regimens.

Since the initiation of this study, approvals for the promising drug class of CPIs continues to burgeon [38]. A recent review indicates that in the classes of PD-1 and PD-L1 inhibitors approved treatment indications span 19 cancer types and 2tissue agnostic conditions (marker-based) [39]. Given the increasing prevalence of use, determination of the cognitive impact of these agents, singly and in combination with other therapies, remains a critical need.

Numerous other factors may contribute to changes in cognitive function for people with non-CNS malignancies, including comorbidities affecting oxygenation levels and pertinent nutritional deficiencies. Notably, elevation in inflammatory cytokine levels is also associated with a constellation of symptoms accompanying changes in cognitive function, sometimes referred to as sickness behavior, namely fatigue, sleep disturbance, anxiety, and depression [17]. Patients experiencing some or all of these may report more significant cognitive changes. We did not control for these potential factors in this small feasibility pilot. The pilot study design did not include the use of neuroimaging to measure any related structural changes in the brain that may accompany treatment with CPIs and would be of interest in future studies.

Future research with a larger sample is needed to address these limitations. Given the number of cancer diagnoses treated with CPIs and the variety of approved drug regimens, formation of a multi-site observational registry would be ideal to obtain the sample size needed for stratification by tumor type and CPI regimen. Collaboration among National Cancer Institute (NCI)-designated Cancer Centers whose parent institutions are also recognized as NACC ADRC centers would leverage existing infrastructure for neuropsychological assessments and access to both blood samples and neuroimaging.

## 5. Conclusions

Results from this small observational pilot indicate that treatment with CPI(s) may have a negative impact on performance for some neurocognitive domains and contribute to the changes in cognitive function reported by individuals diagnosed with cancer. Further investigation is warranted. A multi-site study design may be critical to achieving the necessary power to critically examine the impact of CPI therapy on cognitive function. A potential solution may be the establishment of a multi-site observational registry. Partnerships with National Cancer Institute-designated Cancer Centers, whose parent institutions also are recognized as NACC ADRC centers, may provide the necessary standardization of infrastructure to conduct further investigation. Providers caring for patients receiving CPIs should be aware of this potential effect as they assess patients for treatment-related sequalae.

## Figures and Tables

**Figure 1 cancers-15-01615-f001:**
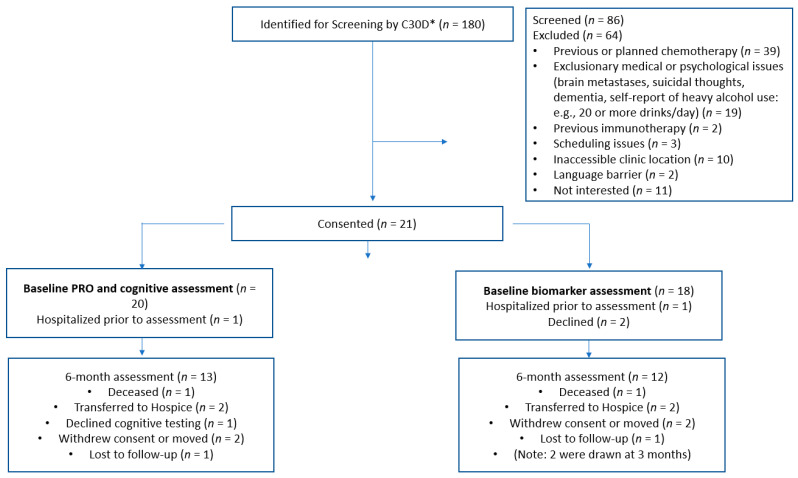
Consort diagram. * Curated Clinical Outcomes Database.

**Figure 2 cancers-15-01615-f002:**
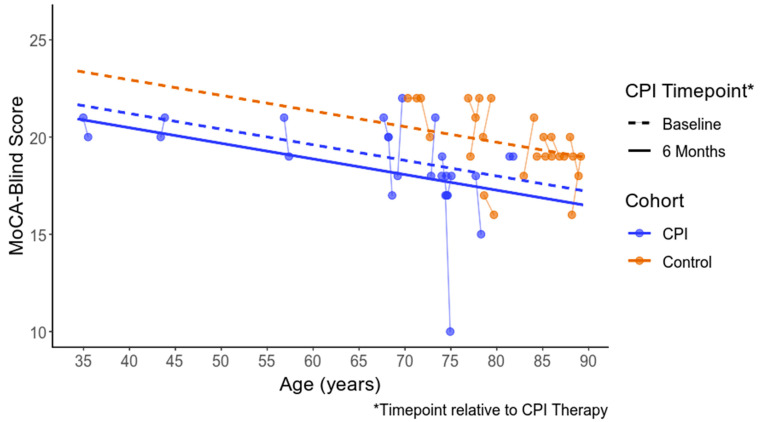
MoCA-Blind scores by cohort and timepoint.

**Table 1 cancers-15-01615-t001:** Neurocognitive assessment tests.

Measure	Domain	Brief Description	Phone Adaptation
Montreal Cognitive Assessment (MoCA)	Global cognitive score for short-term memory recall, visuospatial ability, attention/concentration, working memory, and abstract reasoning	Participants complete a series of tasks including recall of five nouns, a clock-drawing and three-dimensional cube copy test, target detection, serial subtraction, a three-item naming task, similarity description task, and orientation to time and place. Scores range from 0 to 30.	MoCA Blind Assessment excludes clock-drawing and three-dimensional cube copy test
Craft Story 21 Recall (immediate and delayed)	Episodic memory	Participants are read a short story and asked to repeat as much as they can remember immediately upon hearing the story and following a 20 min delay. Verbatim scores range from 0 to 44 correct words. Paraphrase scores range from 0 to 25 correct components.	
Digit Span Forward and Backward	Working memory	Number sequences are presented verbally in ascending order of length. Participants are asked to recall the numbers in both forward and backward sequences. Scored by number of correct trials and longest correct sequence span. Forward span trials range from 0 to 14 with spans of 3–9. Backward span trials range from 0 to 14 with spans of 2–8.	
Trail Making Test (TMT) A	Processing speed	Participants connect 25 circles containing numbers in numerical order as quickly as they can. Score includes number of seconds to complete, number of correct lines, and number of errors.	Oral Trail Making A. Participants orally respond with sequential numbers 1–25.
Trail Making Test (TMT) B	Executive function	Participants are asked to connect 25 circles containing either numbers or letters, alternating numerical with alphabetical order as quickly as they can. Scoring same as above.	Oral Trail Making B. Participants orally respond with alternating sequential numbers and letters (1–12, A-L)
Verbal Fluency (F and L)	Verbal fluency	Participants name as many words as they can in one minute starting with a specified target letter.	
Category Fluency (animals and vegetables)	Category fluency	Participants name as many words as they can in one minute within a specified target category.	
Digit Symbol	Processing speed, attention, visuo-perceptual function, and executive function	Participants are asked to use a key to match the randomly presented numbers 1–9 with easy to draw symbols in a timed test (90 or 120 s, up to 100 items). Scores equal number of correct answers	Omit
Block Design	Visuospatial ability	Participants are asked to use 3-dimensional cubes to replicate a series of up to 14 figures ascending in complexity in a timed test. Scores include the number of correct designs replicated within the time allowed. A total of 4 pts are awarded for each correct figure (maximum score = 48). Time bonuses also may be calculated if desired.	Omit
Stroop Test	Selective attention, inhibition, and processing speed	Participants are presented with a list of words that name common colors (e.g., red, blue, green). The words are printed in colors that differ from the meaning of the word (e.g the word red is written in blue ink). Participants must read the word as listed, not the color in which it is printed. Scores include response time and error rate.	Omit
Letter-Number Sequencing (LNS)	Working memory	Participants are verbally presented with random lists of letters and numbers in ascending length. Participants are asked to repeat the series listing the numbers in numerical order and the letters in alphabetical order.	

**Table 2 cancers-15-01615-t002:** Self-report instruments.

Measure	Description
PROMIS * Cognitive Function 8a	An 8-item Likert-style short form to assess participants perceptions of cognitive problems. Items are ranked from 1 to 4. Raw scores are converted to T-scores and standard error. Higher scores indicate better cognitive function.
PROMIS Cognitive Abilities 8a	As above, but measures participants’ perceptions of cognitive abilities.
Geriatric Depression Scale (GDS)	A 15-item short form, 1 point for each “yes” answer. Higher scores indicate more depression.
Geriatric Anxiety Scale (GAS)	A 10-item Likert style ranking from 0 to 3. Higher scores indicate more anxiety.
NACC ** Functional Assessment Scale	A 10-item Likert-style form ranked 0–3 with higher scores indicating higher levels of dependence in activities of daily living.
Pittsburgh Sleep Quality Index (PSQI)	Narrative and Likert-style instrument measuring 7 components of sleep quality. Yields a global score of sleep quality (0–21). Lower scores indicate better sleep quality
International Physical Activity Questionnaire (IPAQ)	A 7-item measure of low, moderate, and vigorous activity. Yields a metabolic equivalents (MET) total in minutes per week.

* Patient Reported Outcomes Measurement Information System. ** National Alzheimer’s Coordinating Center.

**Table 3 cancers-15-01615-t003:** CPI group demographics baseline vs. month 6.

	Baseline (*N* = 20)	Month 6 (*N* = 13)
**Age (years)**		
Mean (SD)	68.1 (15.3)	66.4 (13.9)
Median [Min, Max]	73.5 [32.0, 88.0]	72.0 [34.0, 81.0]
**Sex**		
Female	8 (40.0%)	5 (38.5%)
Male	12 (60.0%)	8 (61.5%)
**Education (years)**		
Mean (SD)	14.9 (3.03)	15.0 (2.94)
Median [Min, Max]	14.5 [12.0, 20.0]	16.0 [12.0, 20.0]
**Ethnicity**		
Not Hispanic	20 (100%)	13 (100%)
**Race**		
Black or African American	1 (5.0%)	1 (7.7%)
White	19 (95.0%)	12 (92.3%)
**Tumor Type**		
Head and neck squamous cell carcinoma of hard palate	1 (5.0%)	1 (7.7%)
Hepatocellular	2 (10.0%)	2 (15.4%)
Melanoma	12 (60.0%)	7 (53.8%)
Non-small cell lung cancer	1 (5.0%)	0 (0%)
Renal Cell	2 (10.0%)	2 (15.4%)
Squamous cell carcinoma of orbit	1 (5.0%)	0 (0%)
Urothelial	1 (5.0%)	1 (7.7%)
**Number of Visits**		
1	7 (35.0%)	0 (0%)
2	13 (65.0%)	13 (100%)

**Table 4 cancers-15-01615-t004:** ADRC controls vs. CPI group at month 6—demographics.

	Control (*N* = 13)	CPI (*N* = 13)	Overall (*N* = 26)
**Age (years)**			
Mean (SD)	80.8 (6.10)	66.4 (13.9)	73.6 (12.8)
Median [Min, Max]	82.0 [70.0, 88.0]	72.0 [34.0, 81.0]	75.5 [34.0, 88.0]
**Sex**			
Female	8 (61.5%)	5 (38.5%)	13 (50.0%)
Male	5 (38.5%)	8 (61.5%)	13 (50.0%)
**Education (years)**			
Mean (SD)	16.4 (2.22)	15.0 (2.94)	15.7 (2.65)
Median [Min, Max]	16.0 [12.0, 20.0]	16.0 [12.0, 20.0]	16.0 [12.0, 20.0]
**Ethnicity**			
Not Hispanic	13 (100%)	13 (100%)	26 (100%)
**Race**			
Black or African American	2 (15.4%)	1 (7.7%)	3 (11.5%)
White	11 (84.6%)	12 (92.3%)	23 (88.5%)
**Number of Visits**			
2	13 (100%)	13 (100%)	26 (100%)

**Table 5 cancers-15-01615-t005:** Overall scores for self-report instruments—change from baseline estimates.

Test	Estimate	Standard Error	95% Confidence Interval	*p*-Value
PROMIS Cognitive Abilities T-Score	−3.195	2.629	−8.981, 2.592	0.250
PROMIS Cognitive Function T-Score	−3.239	2.789	−9.377, 2.899	0.270
Functional Activities Scale	0.509	1.177	−2.081, 3.099	0.673
Geriatric Anxiety Scale	1.408	1.150	−1.122, 3.939	0.246
Geriatric Depression Scale	0.922	0.763	−0.756, 2.601	0.252
IPAQ MET—minutes/week	1514.886	1062.387	−823.412, 3853.184	0.182
Pittsburgh Sleep Quality Index	0.248	1.015	−1.987, 2.483	0.812

**Table 6 cancers-15-01615-t006:** PSQI scores—change from baseline estimates.

Test	Estimate	Standard Error	95% Confidence Interval	*p*-Value
Daytime Dysfunction Due to Sleepiness	0.128	0.181	−0.271, 0.527	0.494
Duration of Sleep	−0.198	0.285	−0.826, 0.430	0.502
Need Meds to Sleep	−0.033	0.424	−0.966, 0.900	0.940
Overall Sleep Quality *	1.129	0.398	0.520, 2.452	0.737
Sleep Disturbance *	0.933	0.265	0.500, 1.742	0.812
Sleep Efficiency *	0.948	0.357	0.414, 2.172	0.889
Sleep Latency	0.390	0.342	−0.363, 1.143	0.279

* Modeled assuming a Poisson distribution, indicating a multiplicative change from baseline.

**Table 7 cancers-15-01615-t007:** Cognitive scores—change from baseline estimates.

Test	Estimate	Standard Error	95% Confidence Interval	*p*-Value
1a. MoCA-Blind TOTAL RAW SCORE—UNCORRECTED	−0.260	0.831	−2.088, 1.569	0.760
2a. Total Craft story units recalled, verbatim scoring	2.833	1.960	−1.482, 7.148	0.176
2b. Total Craft story units recalled, paraphrase scoring	0.781	1.311	−2.104, 3.665	0.563
3a. Digit symbol forward number of correct trials	0.505	0.460	−0.507, 1.517	0.296
3b. Longest span forward	0.203	0.267	−0.384, 0.790	0.462
4a. Digit symbol backward number of correct trials	0.576	0.542	−0.616, 1.768	0.311
4b. Longest span backward	0.429	0.318	−0.270, 1.128	0.204
5a. Trail Making Test A: Total number of seconds to complete	−0.722	0.860	−2.614, 1.169	0.419
5b. Trail Making Test B: Total number of seconds to complete	−16.855	14.772	−49.369, 15.658	0.278
5b1. Number of commission errors	−1.025	0.632	−2.415, 0.365	0.133
6a. Category Fluency number of animals	−0.751	1.089	−3.148, 1.646	0.505
6b. Category Fluency number of vegetables	1.310	0.997	−0.885, 3.505	0.216
7a. Verbal Fluency number of correct F-words generated in 1 min	0.818	1.077	−1.552, 3.187	0.464
7b. Verbal Fluency number of F-words repeated in 1 min	0.231	0.371	−0.586, 1.048	0.547
7d. Verbal Fluency number of correct L-words generated in 1 min	0.632	1.000	−1.569, 2.832	0.541
7g. Verbal Fluency total number of correct F-words and L-words *	1.072	0.086	0.898, 1.280	0.405
7h. Verbal Fluency total number of F-words and L-words repetition errors *	1.343	0.454	0.638, 2.826	0.401
7i. Verbal Fluency total number of non-F/L words and rule violation errors *	0.099	0.103	0.010, 0.972	0.048
8a. Total Craft story delayed units recalled, verbatim scoring	1.374	1.765	−2.511, 5.258	0.453
8b. Total story delayed units recalled, paraphrase scoring	−0.261	1.055	−2.583, 2.062	0.809
9a. Letter number sequencing	1.791	1.249	−0.991, 4.574	0.182

* Modeled assuming a Poisson distribution, indicating a multiplicative change from baseline value. Note: Change from baseline was not estimated for the following tests due to low variability in the responses: 5a1. Number of commission errors; 5a2. Number of correct lines; 5b2. Number of correct lines; 7c. Number of F-words and rule violation errors in 1 min; 7e. Number of L-words repeated in 1 min; 7f. Number of non-L-words and rule violation errors in 1 min; 8c. Delay time (minutes); 8d. Cue (boy) needed.

**Table 8 cancers-15-01615-t008:** Estimates for differences in MOCA-Blind scores.

Cohort/Timepoint	Estimate	95% Confidence Interval	*p*-Value
CPI Group After Treatment vs. Before Treatment	−0.730	−2.085, 0.624	0.277
CPI Group Before Treatment vs. ADRC Controls	−1.735	−3.591, 0.122	0.066
CPI Group After Treatment vs. ADRC Controls	−2.465	−4.304, −0.627	0.011

**Table 9 cancers-15-01615-t009:** Biomarkers—change from baseline estimates.

Biomarker	Estimate	95% Confidence Interval	*p*-Value
BDNF (pg/mL)	1.115	0.854, 1.454	0.385
FGF2 (pg/mL)	1.470	0.724, 2.985	0.250
IFG-1 (pg/mL)	1.180	0.984, 1.415	0.069
IFNg (pg/mL)	0.650	0.245, 1.723	0.331
IL-1a (pg/mL)	1.106	0.856, 1.430	0.299
IL-1b (pg/mL)	0.807	0.600, 1.087	0.139
IL-2 (pg/mL)	0.809	0.608, 1.077	0.128
IL-6 (pg/mL)	0.984	0.575, 1.683	0.947
TNFa (pg/mL)	1.130	0.823, 1.552	0.411
VEGF (pg/mL)	1.030	0.675, 1.572	0.875

Biomarkers were modeled assuming a log-normal distribution. Therefore, the estimates indicate a multiplicative change from baseline.

**Table 10 cancers-15-01615-t010:** Pearson correlations for change from baseline between biomarkers and cognitive scores.

Test (pg/mL)	IFN	IL-1a	IL-1b	IL-2	TNFa	VEGF	IL-6	FGF2	IGF-1	BDNF
1a. MoCA-Blind TOTAL RAW SCORE—UNCORRECTED										
	0.064	−0.047	0.091	0.098	−0.119	0.388	−0.223	0.076	0.219	0.579
2a. Total Craft story units recalled, verbatim scoring										
	−0.811 *	−0.946	−0.820 *	−0.820 *	−0.412	−0.885 *	−0.534	−0.716 *	0.217	−0.423
2b. Total Craft story units recalled, paraphrase scoring										
	−0.835 *	−0.945	−0.799 *	−0.798 *	−0.458	−0.910 *	−0.519	−0.710 *	0.217	−0.473
3a. Digit symbol forward number of correct trials										
	−0.645	−0.909	−0.524	−0.520	−0.213	−0.467	−0.475	−0.400	0.312	−0.571
3b. Longest span forward										
	−0.538	−0.845	−0.468	−0.466	−0.110	−0.498	−0.456	−0.496	0.554	−0.462
4a. Digit symbol backward number of correct trials										
	0.344	0.576	0.314	0.317	0.167	0.716	0.166	0.273	−0.004	0.548
4b. Longest span backward										
	0.367	0.576	0.373	0.379	0.291	0.862 *	0.282	0.352	−0.219	0.660
5a. Trail Making Test A: Total number of seconds to complete										
	−0.175	−0.685	−0.133	−0.126	0.037	−0.062	−0.053	−0.313	−0.222	−0.053
5b. Trail Making Test B: Total number of seconds to complete										
	−0.192	−0.977 *	−0.209	−0.209	−0.027	−0.352	−0.165	−0.424	0.412	−0.296
6a. Category Fluency number of animals										
	0.123	0.012	0.086	0.086	0.322	0.021	0.168	0.174	−0.133	0.315
6b. Category Fluency number of vegetables										
	−0.749	−0.750	−0.334	−0.339	−0.405	−0.719	−0.372	−0.314	0.241	0.415
7a. Verbal Fluency total number of correct F-words generated in 1 min										
	−0.212	−0.830	−0.181	−0.177	0.031	−0.086	−0.064	−0.073	−0.150	−0.033
7b. Verbal Fluency total number of F-words repeated in 1 min										
	−0.055	−0.331	−0.025	−0.022	0.304	0.157	−0.131	0.006	0.232	−0.074
7d. Verbal Fluency total number of correct L-words generated in 1 min										
	−0.375	−0.560	−0.385	−0.391	−0.027	−0.531	−0.141	−0.169	0.050	−0.391
7g. Verbal Fluency total number of correct F-words and L-words										
	−0.336	−0.746	−0.334	−0.336	−0.006	−0.414	−0.127	−0.140	−0.042	−0.244
7h. Verbal Fluency total number of F-words and L-words repetition errors										
	−0.078	−0.347	−0.047	−0.043	0.230	0.157	−0.248	−0.116	0.424	−0.086
7i. Verbal Fluency total number of non-F/L words and rule violation errors										
	0.081	0.331	0.005	0.000	−0.028	−0.161	0.032	−0.221	0.278	0.121
8a. Total Craft story delayed units recalled, verbatim scoring										
	−0.420	−0.988 *	−0.378	−0.370	−0.034	−0.451	−0.152	−0.284	−0.195	−0.040
8b. Total Craft story delayed units recalled, paraphrase scoring										
	−0.571	−0.919	−0.582	−0.580	−0.309	−0.705	−0.298	−0.472	−0.083	−0.053
8c. Delay time (minutes)										
	0.162	0.295	0.184	0.192	0.158	0.222	0.242	0.114	−0.360	−0.287
13. Letter number sequencing										
	0.164	−0.070	0.135	0.132	0.296	0.215	−0.159	−0.035	0.707 *	−0.068

* Correlation is significant (*p*-value < 0.05). Note: Correlations were not estimated for the following tests due to low variability in the responses: 5a1. Number of commission errors; 5a2. Number of correct lines; 5b1. Number of commission errors.

## Data Availability

The data presented in this study are available on request from the corresponding author. The data are not publicly available due to privacy restrictions.

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
