# Peer review of "First-Line Immunotherapy with Check-Point Inhibitors: Prospective Assessment of Cognitive Function"

_cancers, 2023, doi:10.3390/cancers15051615_

Round 1

Reviewer 1 Report

Myers JS et al. evaluated the effect of the inhibitors for the immune checkpoints as first-line immunotherapy on cognitive function. It has been demonstrated that chemotherapy is toxic and induces severe side effects, including impairment of cognitive function. Also, immunotherapy is toxic since checkpoint inhibitors are associated with immune-related adverse events. However, little is known about the toxicities of these checkpoint inhibitors on cognitive function. Therefore, this study shed some light on this question. The authors found that checkpoint inhibitors may negatively impact some neurocognitive domains; thus, further research must be done to avoid the risk for cancer patients.

The number of patients is limited, but the study has value.

Some minor points should be addressed:

In the results section, table 10 includes many numbers that are not easy to follow.

Please, discuss further the relevance of the inflammatory cytokines and their relationship with neurotoxicity.

The authors should discuss if the tests used are correct for this study. Are there any others they could have included?

Author Response

We are most grateful for the thoughtful and thorough review of our manuscript, First-Line Immunotherapy with Check-point Inhibitors: Prospective Assessment of Cognitive Function that was submitted for the special issue of Cancers. Addressing the concerns raised by the review has strengthened our manuscript.

Reviewer 1:

In the results section, table 10 includes many numbers that are not easy to follow.-

  • Response- Part of the issue here may be the required formatting for the journal. Pending permission from the editorial team, we left justified the internal table headings. We also moved the pg/mL descriptor over to the Test Heading to declutter the primary heading a bit. We hope this makes the numbers less dense and easier to read.

Please, discuss further the relevance of the inflammatory cytokines and their relationship with neurotoxicity.

  • Response- additional detail is provided regarding cytokine production in the peripheral blood and central nervous system and resulting injury to neuroprogenitor cells.

The authors should discuss if the tests used are correct for this study. Are there any others they could have included?

  • Response- We added further detail regarding the use of standardized neuropsychological tests utilized by the National Alzheimer’s Coordinating Center for the Uniform Data set in addition to a few additional core tests utilized specifically for the University of Kansas Alzheimer’s Disease Research Center. The selected tests addressed measurement of the target cognitive domains, namely attention/concentration, working memory, processing speed and visuospatial ability.

Reviewer 2 Report

The manuscript "First-Line Immunotherapy with Check-point Inhibitors:Prospective Assessment of Cognitive Function" aims to understand the cognitive effects of immunotherapy with Check-point Inhibitors in cancer patients. The manuscript is interesting, however there are some questions to be addressed:

- Please provide the rational for the control group. Why the authors decided to have a control group with cognitie disfunction caused by Alzheimer´s disease.

- Please provide the inclusion and exclusion criteria.

- The increase levels of IFN, Il-1B, IL-2, FGFR2 and VEGF that the authors claim that is associated witg cognitive disfunction can be only correlated with tumour biology. How the authors can be sure that is associated with cognite disfunction, rather than inflammation or tumour´s biology? The increase levels can be soociate with the release of inflammatory compounds from the tumor. Please provide an explanation.

- The number of patients is very small and the p values in almost all studies are not statistical significant. Maybe merge some categories will improve the p values.

Author Response

We are most grateful for the thoughtful and thorough review of our manuscript, First-Line Immunotherapy with Check-point Inhibitors: Prospective Assessment of Cognitive Function that was submitted for the special issue of Cancers. Addressing the concerns raised by the review has strengthened our manuscript.

Reviewer 2:

- Please provide the rationale for the control group. Why the authors decided to have a control group with cognitive disfunction caused by Alzheimer´s disease.

  • Response- We added further detail to clarify that the control group was comprised of cognitively intact individuals who are assessed annually as participants in the National Alzheimer’s Coordinating Center Uniform Data set. These cognitively intact participants’ neuropsychological test scores are used for comparison purposes and provided us with age-matched controls with which to assess/compare the Checkpoint-Inhibitor Group participants’ scores.

- Please provide the inclusion and exclusion criteria.

  • Response- Inclusion exclusion criteria for the cognitively intact controls was added as requested.

- The increase levels of IFN, Il-1B, IL-2, FGFR2 and VEGF that the authors claim that is associated with cognitive disfunction can be only correlated with tumour biology. How the authors can be sure that is associated with cognitive disfunction, rather than inflammation or tumour´s biology? The increase levels can be associated with the release of inflammatory compounds from the tumor. Please provide an explanation.

  • Response- The Reviewer is correct that many variables may cause an increase in cytokines, including both the tumor biology and body’s inflammatory response as well as an inflammatory response due to the cancer treatment. Our aim for this small feasibility pilot was simply to garner preliminary data about change in cognitive function for individuals on 1st-line treatment with CPIs and see whether there was a correlation with systemic markers of inflammation or neuroprotection. Our study was not designed or powered to confirm a relationship between CPIs, inflammation, and cognitive performance.

- The number of patients is very small and the p values in almost all studies are not statistically significant. Maybe merge some categories will improve the p values.

  • Response- We agree that this is a very small sample. However, that was by intention as the primary aims of the pilot were to demonstrate feasibility for recruitment, retention, and data collection from this population so that we can inform a larger study that is powered to show significant change in cognitive function. We were excited to see preliminary results to support the plan for further investigation.

Reviewer 3 Report

 The proposed paper is prepared very well, with excellent quality of presentation of material and methods and results. It was a pleasure to read it.

The research question is clearly outlined. The authors stated clearly what study found and how they did it. The title is informative and relevant.

The references are relevant and recent. The cited sources are referenced correctly. Appropriate and key studies are included.

The process of selection of the subjects was clear. The variables are well defined and measured appropriately. The study methods are valid and reliable. There are enough details provided in order to replicate the study.

The data is presented in an appropriate way. The text in the results add to the data and it is not repetitive. Statistically significant results are clear. It is clear which results are with practical meaning. Results are discussed from different angles and placed into context without being overinterpreted.

The conclusions answer the aim of the study. The conclusions are supported by references and own results.

The limitations of the study are not fatal, but they are opportunities to inform future research.

Specific comments on weaknesses of the article and what could be improved:

Major points  - none        

Minor points

1.           Figure 1 could be more easy to read if you remove the outer frame

2.           How can you discuss the correlations between biomarkers and  neurocognitive performance (described in 337-343), and then mention again in discussion (378-382). What could be the biological explanation/connection for such correlation? This statement "A potential explanation may be that VEGF has the potential for both inflam- 382 matory and anti-inflammatory effects [36]." should be extended a bit.

3.          Based on your results and conclusions, do you think that follow up,  registering and active monitoring of patients on CPI therapy are enough measures, that they can be recommended as routine?

Author Response

We are most grateful for the thoughtful and thorough review of our manuscript, First-Line Immunotherapy with Check-point Inhibitors: Prospective Assessment of Cognitive Function that was submitted for the special issue of Cancers. Addressing the concerns raised by the review has strengthened our manuscript.

  •  

Reviewer 3:

Minor points

  1.          Figure 1 could be more easy to read if you remove the outer frame-
  • Response: The outer frame has been removed as requested.
  1.          How can you discuss the correlations between biomarkers and neurocognitive performance (described in 337-343), and then mention again in discussion (378-382). What could be the biological explanation/connection for such correlation? This statement "A potential explanation may be that VEGF has the potential for both inflammatory and anti-inflammatory effects [36]." should be extended a bit.
  • Response: Further detail has been added regarding the role of VEGF in both neuroprotection and inflammation in the Discussion section. We also added a bit more explanation of the relationship between cytokines and neuroprogenitor cell injury.
  1.         Based on your results and conclusions, do you think that follow up registering and active monitoring of patients on CPI therapy are enough measures, that they can be recommended as routine?
  • Response: We cannot base recommendations for routine or standard of care based on the preliminary results of this small feasibility pilot. We can only use these results to support the need for further, well-powered investigation to truly determine the impact of CPI’s on cognitive function longitudinally. And, we do suggest that clinician’s be aware of the potential cognitive effect of CPIs as they assess patients during and following treatment.

Reviewer 4 Report

Thank you for asking me to review the manuscript by Myers et al. titled “First-Line Immunotherapy with Check-point Inhibitors: Prospective Assessment of Cognitive Function”. In this study, the authors have conducted a small pilot study to test the impact of first-line immune checkpoint therapy on cognitive abilities on older adults by comparing changes at baseline and post 6 months and comparing the results with age-matched controls from the Alzheimer’s Disease Research Center (ADRC). Additionally, the purpose of the study was to evaluate the feasibility of conducting such studies associated with recruitment and testing. Further, the authors have also correlated changes in inflammatory biomarkers that have been shown to be associated with neurocognition.

Overall, these studies are important as Checkpoint inhibitors (CPIs) are becoming to standard of care cancer therapy for many cancer types and not much is known how this therapy impacts quality of life including cognitive abilities. Therefore, developing database and studies that include such information when recruiting patients into clinical trials might be vital. However, there many caveats associated with the study design and conclusions made by the authors that would need to be assessed carefully to make any significant conclusions. My specific comments are as follows:

Major comments:

1)     The authors highlight that there was no difference in the inflammatory mediators between baseline and 6 months post therapy with CPI (line 281). In that case, do we know if there is a significant increase in these inflammatory mediators in these specific patients that be correlated with the neurocognitive changes? Have the authors compared expressionof the inflammatory mediator expression with healthy controls to ascertain changes in the markers? This will be important to make any relevant conclusions in the downstream analysis.  

2)     Comparative analysis has been performed between 6 months post CPI therapy and controls

Alzheimer’s Disease Research Center (ADRC)’s 12 month performance. The authors do not comment on whether this comparison is relevant or what assumptions/ caveats are associated with such comparisons. This should be included in the main text/discussion. Additionally, a proper 1:1 12 month post CPI therapy with controls would need to be carried out for more appropriate conclusions. 

3)     As also mentioned by the authors, irAEs are higher in anti-CTLA-4 treated patients but most patients used in this study have been treated with anti-PD-1 (nivo/ pembro). Therefore, comparative expression of the inflammatory mediators compared to controls (mentioned in point 1) as well more recruitment of patients treated with anti-CTLA-4 or both anti-CTLA-4 and anti-PD-1 would be important to make better correlations.

4)     60% of the patients analyzed here had melanoma and there are 1 or 2 patients each with other tumor types. Since different tumors secrete different levels of inflammatory mediators this could induce variability or impact the differences between control and test groups. As majority had melanoma, the authors should also perform another analysis of just comparing the data of these patients with the controls. This might provide more insight for a specific tumor type with less confounding variables.

5)     Is peripheral blood the best sample for this analysis? Wouldn’t assessing the cerebrospinal fluid be better to check for level of the mediators that cross the Blood brain barrier?

6)     A big caveat in this study is that a simple 1:1 correlation has been carried out by correlating expression of markers with neurocognitive abilities? However, there could be so many factors that could led to changes in cognitive abilities. For eg. B-12 deficiency is high in cancer patients and could led to neurocognitive effects. These patients may also have increase in inflammatory mediators that are probably not associated with these effects. These points need to be highlighted in the discussion section.

7)     The authors should also consider variables such as immunotherapy related structural changes in the brain (and not only increase in inflammatory mediators) via MRI that could be associated with neuro impairment.

Minor comments:

1) typo – line 458- reference 13 in the word “recommendation”

Author Response

We are most grateful for the thoughtful and thorough review of our manuscript, First-Line Immunotherapy with Check-point Inhibitors: Prospective Assessment of Cognitive Function that was submitted for the special issue of Cancers. Addressing the concerns raised by the review has strengthened our manuscript.

REVIEWER 4:

Major comments:

  • The authors highlight that there was no difference in the inflammatory mediators between baseline and 6 months post therapy with CPI (line 281). In that case, do we know if there is a significant increase in these inflammatory mediators in these specific patients that be correlated with the neurocognitive changes? Have the authors compared expression of the inflammatory mediator expression with healthy controls to ascertain changes in the markers? This will be important to make any relevant conclusions in the downstream analysis.  
  • Response: The primary purpose of the pilot study was feasibility. Given funding and timing restrictions we only collected and analyzed blood samples from the CPI group. So, thus far no comparisons for biomarker levels have been made with the cognitively intact controls. The findings from this small pilot must be considered very preliminary, and as such we can only report descriptive statistics to inform future research. We added this information to the limitations noted in the Discussion section.

2)     Comparative analysis has been performed between 6 months post CPI therapy and controls

Alzheimer’s Disease Research Center (ADRC)’s 12 month performance. The authors do not comment on whether this comparison is relevant or what assumptions/ caveats are associated with such comparisons. This should be included in the main text/discussion. Additionally, a proper 1:1 12 month post CPI therapy with controls would need to be carried out for more appropriate conclusions. 

  • Response: We added further detail regarding this study limitation and plans for addressing in future studies.
  • As also mentioned by the authors, irAEs are higher in anti-CTLA-4 treated patients but most patients used in this study have been treated with anti-PD-1 (nivo/ pembro). Therefore, comparative expression of the inflammatory mediators compared to controls(mentioned in point 1) as well more recruitment of patients treated with anti-CTLA-4 or both anti-CTLA-4 and anti-PD-1 would be important to make better correlations.

  • Response: We added this limitation to the Discussion section. Plans for future multi-site observational registry would allow sufficient power to stratify by CPI regimen.

  • 60% of the patients analyzed here had melanoma and there are 1 or 2 patients each with other tumor types. Since different tumors secrete different levels of inflammatory mediators this could induce variability or impact the differences between control and test groups. As majority had melanoma, the authors should also perform another analysis of just comparing the data of these patients with the controls. This might provide more insight for a specific tumor type with less confounding variables.

  • Response: We conducted a post facto analyses for the melanoma subgroup. Results mirrored that of the full CPI group compared to the controls. We added these results to the manuscript. Plans for future multi-site observational registry will allow sufficient power to stratify by disease type.
  • Is peripheral blood the best sample for this analysis? Wouldn’t assessing the cerebrospinal fluid be better to check for level of the mediators that cross the Blood brain barrier?

  • Response: One could argue that CSF analyses may be ideal, but since cytokines can cross the BBB in addition to production within the CNS, blood sampling is a practical, affordable, and less invasive method to determine biomarker levels. Of note, CSF and blood sampling are optional components of the consent obtained from the cognitively intact participants in the ADRC cohort. Use of CSF could be considered in the future.

  • A big caveat in this study is that a simple 1:1 correlation has been carried out by correlating expression of markers with neurocognitive abilities? However, there could be so many factors that could led to changes in cognitive abilities. For eg. B-12 deficiency is high in cancer patients and could led to neurocognitive effects. These patients may also have increase in inflammatory mediators that are probably not associated with these effects. These points need to be highlighted in the discussion section.

  • Response: We added further detail regarding study limitations. However, we did not specifically “call out” vitamin B-12. Our understanding is B-12 deficiencies commonly are found for patients with gastric cancer, for those who have had gastrectomy, or those receiving chemotherapy. We included a more broadly worded limitation statement i.e.. referring to “pertinent nutritional deficiencies”. We also added some detail to address the association of elevated inflammatory cytokine levels sickness behavior, namely fatigue, sleep disturbance, anxiety, depression. Patients experiencing some or all of these may report more significant cognitive changes.
  • The authors should also consider variables such as immunotherapy related structural changes in the brain (and not only increase in inflammatory mediators) via MRI that could be associated with neuro impairment.

  • Response: We noted that an important consideration for measurement in the future multi-site observational registry is neuroimaging to capture related structural changes in the brain.

Minor comments:

  • typo – line 458- reference 13 in the word “recommendation”
  • Response: typo has been corrected.

Round 2

Reviewer 4 Report

The authors have responded to my questions with satisfaction. 

On a separate point, they have performed the analysis for melanoma patients as requested and have provided the p values in the manuscript, however, the actual table of results is missing and should be included as a supplemental figure if not in the main text.